# Olanzapine Induced Dysmetabolic Changes Involving Tissue Chromium Mobilization in Female Rats

**DOI:** 10.3390/ijms20030640

**Published:** 2019-02-01

**Authors:** Ching-Ping Yang, Ya-Yu Wang, Shih-Yi Lin, Yi-Jheng Hong, Keng-Ying Liao, Sheng-Kuo Hsieh, Ping-Ho Pan, Chun-Jung Chen, Wen-Ying Chen

**Affiliations:** 1Department of Medical Research, Taichung Veterans General Hospital, Taichung 407, Taiwan; milkygp@gmail.com (C.-P.Y.); cjchen@vghtc.gov.tw (C.-J.C.); 2Department of Family Medicine, Taichung Veterans General Hospital, Taichung 407, Taiwan; yywang@vghtc.gov.tw; 3Center for Geriatrics and Gerontology, Taichung Veterans General Hospital, Taichung 407, Taiwan; sylin@vghtc.gov.tw; 4Department of Veterinary Medicine, National Chung Hsing University, Taichung 402, Taiwan; d1245877@gmail.com (Y.-J.H.); emily93211@hotmail.com (K.-Y.L.); t6395@ms.sltung.com.tw (P.-H.P.); 5Graduate Institute of Biotechnology, National Chung Hsing University, Taichung 402, Taiwan; vincent760424@yahoo.com.tw; 6Department of Pediatrics, Tungs’ Taichung Metro Harbor Hospital, Taichung 435, Taiwan; 7Department of Medical Laboratory Science and Biotechnology, China Medical University, Taichung 447, Taiwan

**Keywords:** chromium, hyperglycemia, obesity, olanzapine

## Abstract

Atypical antipsychotics, such as olanzapine, are commonly prescribed to patients with schizophrenic symptoms and other psychiatric disorders. However, weight gain and metabolic disturbance cause adverse effects, impair patient compliance and limit clinical utility. Thus, a better understanding of treatment-acquired adverse effects and identification of targets for therapeutic intervention are believed to offer more clinical benefits for patients with schizophrenia. Beyond its nutritional effects, studies have indicated that supplementation of chromium brings about beneficial outcomes against numerous metabolic disorders. In this study, we investigated whether olanzapine-induced weight gain and metabolic disturbance involved chromium dynamic mobilization in a female Sprague-Dawley rat model, and whether a dietary supplement of chromium improved olanzapine-acquired adverse effects. Olanzapine medicated rats experienced weight gain and adiposity, as well as the development of hyperglycemia, hyperinsulinemia, insulin resistance, hyperlipidemia, and inflammation. The olanzapine-induced metabolic disturbance was accompanied by a decrease in hepatic Akt and AMP-activated Protein Kinase (AMPK) actions, as well as an increase in serum interleukin-6 (IL-6), along with tissue chromium depletion. A daily intake of chromium supplements increased tissue chromium levels and thermogenic uncoupling protein-1 (UCP-1) expression in white adipose tissues, as well as improved both post-olanzapine weight gain and metabolic disturbance. Our findings suggest that olanzapine medicated rats showed a disturbance of tissue chromium homeostasis by inducing tissue depletion and urinary excretion. This loss may be an alternative mechanism responsible for olanzapine-induced weight gain and metabolic disturbance.

## 1. Introduction

Schizophrenia is a chronic and debilitating mental illness. Its long-lasting health, social, and financial burdens on humans are rising, due to its high global prevalence. Although there are several options regarding antipsychotic medicine, partial patient compliance is associated with increased morbidity and mortality due to adverse effects [1,2]. Recently, atypical antipsychotics or second-generation antipsychotics have replaced conventional antipsychotics as the commonly prescribed medication for acute or maintenance therapy of schizophrenia due to their superior therapeutic uses and reduced risk of extrapyramidal sequelae. Unfortunately, the clinical utility of atypical antipsychotics is limited by concerns regarding their side effects of weight gain and metabolic disturbance [3,4,5]. The development of weight gain and metabolic adverse effects could further lead to increased morbidity and mortality, along with poor compliance to antipsychotic medications for patients with schizophrenia. Therefore, a better understanding of atypical antipsychotics-accompanied weight gain and metabolic disturbance, as well as the identification of prevention and/or treatment options is believed to have more clinical benefits for patients with schizophrenia.

Clinical findings suggest that many atypical antipsychotics, particularly olanzapine and clozapine, are associated with weight gain, glucose abnormality, insulin resistance, and hyperlipidemia which can increase the risk of developing cancer, along with metabolic and cardiovascular diseases. Data from an epidemiological study reveal that the incidence of metabolic disturbance after atypical antipsychotics treatment is to be 20–60%, with the incidence being twice that of the general population [4,6,7,8,9]. The proposed pharmacological activities of atypical antipsychotics include acting on the dopaminergic D2 receptor, histaminergic H1 receptor, serotonergic 2A/2C receptor, muscarinic M3 receptor, and ghrelin receptor. Treatments with atypical antipsychotics have been reported to increase the circulating level of ghrelin, expression of the ghrelin receptor and cytokines in the hypothalamus, fat deposition, and macrophage infiltration and cytokine expression in white adipose tissues, as well as induce atrophy in brown adipose tissues. Thus, the central and peripheral actions are believed to attribute to weight gain and metabolic disturbance caused by atypical antipsychotics [10,11,12,13,14,15,16,17,18,19]. Accumulating evidence indicates that the histamine H1 receptor agonist, endocannabinoid CB1 receptor antagonist, and antiepileptic zonisamide improve olanzapine-induced weight gain and metabolic disturbance [16,20,21,22]. Moreover, metabolism-improving agents, such as metformin, liraglutide, berberine, green tea, and melatonin, show beneficial effects as well [23,24,25,26,27,28]. These phenomena highlight the feasibility of metabolic regulators’ ability to combat atypical antipsychotics-accompanied weight gain and metabolic disturbance.

Trace elements are important to the homeostatic activities of living cells. An excess or deficiency of trace elements leads to an impaired metabolism and the development of metabolic syndrome. Among the trace elements, trivalent chromium is an important element for normal carbohydrate, lipid, and protein metabolism [29]. Chromium deficiency or depletion is frequently diagnosed in numerous metabolic diseases and chromium supplements improve metabolic disturbance and disease pathogenesis [23,30,31,32,33,34]. Currently, the potential involvement of chromium mobilization in atypical antipsychotics-accompanied weight gain and metabolic disturbance still remains unknown.

Our previous studies have found that cerebral ischemia causes tissue chromium mobilization, hyperglycemia, hyperinsulinemia, and insulin resistance. A dietary supplement with chromium improved post-stroke brain injury, cholestatic liver injury, nonalcoholic fatty liver disease, and obesity-impaired insulin signaling [23,31,32,33]. Given its well-known biological activity in regulating glucose and lipid metabolism, we hypothesized that chromium mobilization may play a role in atypical antipsychotics-accompanied weight gain and metabolic disturbance. To extend the scope of the relevant studies and get insight into atypical antipsychotics-associated adverse effects, the aim of the present study was to investigate whether olanzapine causes chromium mobilization, and to determine its potential association with atypical antipsychotics-accompanied weight gain and metabolic disturbance in Sprague-Dawley rats. Moreover, whether chromium supplements bring about beneficial effects against atypical antipsychotics-accompanied weight gain and metabolic disturbance was another focus of this study.

## 2. Results

### 2.1. Olanzapine Caused Weight Gain and Adiposity

In female Sprague-Dawley rats, the twice daily administration of olanzapine at a dose of 2 mg/kg caused an incremental increase in both body weight (Figure 1A) and average body weight gain (P 1B), beginning at days 7 and 5 until day 14, respectively. Olanzapine medicated rats showed a higher daily food intake (Figure 1C), higher white adipose mass (including inguinal, periovarian, mesentery, and perirenal fat) (Figure 1D), and increased liver mass (Figure 1E). However, the difference between gastrocnemius mass was not reaching statistical significance (Figure 1F). A histological examination revealed findings that showed olanzapine medicated rats displayed a slight deposition of lipid droplets in their livers and enlarged adipocytes in their perirenal fat (Figure 1G). These findings suggest that olanzapine treatment has the ability to induce weight gain and adiposity.

### 2.2. Olanzapine Caused Metabolic Disturbance

To extend the histological findings in livers, several metabolic parameters were examined. Serum biochemical analyses showed a negligible difference of aspartate aminotransferase (AST) (Figure 2A) and alanine aminotransferase (ALT) (Figure 2B) between the groups. However, an increment in serum triglycerides (Figure 2C), total cholesterol (Figure 2D), fasting glucose (Figure 2E), insulin (Figure 2F), and Homeostasis Model Assessment-Insulin Resistance (HOMA-IR) (Figure 2G) was found in olanzapine medicated rats. Paralleled Western blotting analyses showed a reduction of Akt, AMP-activated Protein Kinase (AMPK), and acetyl-CoA carboxylase (ACC) phosphorylation in the hepatic tissues of olanzapine medicated rats (Figure 2H). The aforementioned findings suggest an induction of impairment in glucose and lipid metabolism after olanzapine treatment, at least in those involving interference in hepatic insulin action.

### 2.3. Olanzapine Caused Tissue Chromium Mobilization

Chromium is an important element for maintaining glucose and lipid homeostasis, as well as insulin action [29]. It is also valuable to know whether olanzapine treatment causes chromium mobilization. In comparison with the control rats, the chromium levels in the livers, gastrocnemius muscles, white adipose tissues, and serum have decreased, while its level in the urine was increased in olanzapine medicated rats (Figure 3). These results indicate that olanzapine treatment in rats causes chromium tissue depletion and urinary excretion.

### 2.4. Chromium Improved Olanzapine-Induced Weight Gain and Metabolic Disturbance

Since chromium depletion correlated well with olanzapine treatment-induced weight gain and metabolic disturbance, consequent experiments were performed to investigate the potential benefits of chromium supplementation. Daily chromium supplementation for two weeks elevated tissue chromium levels in olanzapine medicated rats, while having a negligible effect in normal rats (Figure 4). The supplementation of chromium in olanzapine medicated rats decreased body weight (Figure 5A), liver mass (Figure 5C), and white adipose mass (Figure 5D). However, the amount of food intake was not changed (Figure 5B). Likewise, hepatic lipid droplet deposition and enlarged adipocytes in olanzapine medicated rats were improved by chromium supplement intake (Figure 5E). In parallel with the histological findings, data of the biochemical study revealed that chromium significantly alleviated the accumulation of hepatic triglycerides caused by olanzapine. However, there was no significant change in hepatic cholesterol (Figure 6A). A reduction in serum triglycerides, total cholesterol (Figure 6B), fasting glucose (Figure 6C), insulin (Figure 6D), and HOMA-IR (Figure 6E) was found in chromium-supplemented olanzapine medicated rats. To further explore the improved results of chromium supplementation against olanzapine-induced weight gain and metabolic disturbance, experiments were performed to investigate insulin regulatory downstream effectors in livers and serum. Results of Western blotting in liver tissues are shown in Figure 6F. There was a decreased expression of Akt phosphorylation, AMPK phosphorylation, and ACC phosphorylation in the olanzapine group, when compared with the control group. These biochemical alterations after olanzapine treatment were attenuated by chromium supplementation (Figure 6F). Besides, olanzapine group increased serum IL-6 level and that increment was alleviated by chromium (Figure 6G). These findings suggest that olanzapine-induced weight gain and metabolic disturbance were accompanied by chromium depletion along with impaired hepatic insulin and AMPK signaling, with chromium supplementation showing an improving effect.

### 2.5. Chromium Increased Thermogenic Pncoupling protein-1 (UCP-1) Expression in White Adipose Tissues

Apart from being a representative marker of beige/brite or brown adipocytes, thermogenic UCP-1 in white adipose tissues represents an emerging molecule in the regulation of energy metabolism and adipose inflammation [35]. Although there was no significant difference of UCP-1 protein expression in white adipose tissues of olanzapine group, chromium showed promoting effect on UCP-1 protein expression and further augmented UCP-1 expression in olanzapine-medicated rats (Figure 7). That is, increased thermogenesis in white adipose tissues might contribute to chromium-improved metabolic disturbance.

## 3. Discussion

Atypical antipsychotics-induced weight gain and metabolic disturbance are not only risk factors for metabolic and cardiovascular diseases leading to higher morbidity and mortality, but their adverse effects compromise a patient’s compliance with antipsychotic medication. Therefore, the identification and development of potential adjunctive medications to prevent and/or treat weight gain and metabolic disturbance induced by atypical antipsychotics isare actively ongoing. A gender difference is generally admitted regarding the obesogenic and diabetogenic adverse effects of olanzapine. Oppositely, clinical and experimental findings indicate that the biological activity of chromium is independent of gender difference [32,36,37]. Using a female Sprague-Dawley rat model, the present study demonstrated that the olanzapine medicated rats experienced weight gain, particularly in the liver and white adipose tissues, and developed hyperlipidemia, hyperglycemia, hyperinsulinemia, and insulin resistance. Olanzapine-induced weight gain and metabolic disturbance were accompanied by decreased hepatic Akt and AMPK phosphorylation along with increased serum IL-6. The present study first reported that olanzapine medicated rats showed a disturbance of tissue chromium homeostasis by inducing tissue depletion and increasing urinary excretion. Daily supplementation studies further suggested that restoration of tissue chromium levels improved olanzapine-induced weight gain and metabolic disturbance. Findings of the current study indicate that tissue chromium mobilization and urinary excretion may be one mechanism which underlies olanzapine-induced weight gain and metabolic disturbance.

Many studies have investigated the metabolic adverse effects of olanzapine via rodent models by focusing on its action on the central hypothalamus, peripheral livers and adipose tissues. Its impact on hypothalamus orexigenic/anorexigenic peptides and inflammation leads to hyperphagia and decreased energy expenditure [10,14,17,20,21]. Olanzapine promotes adipocyte differentiation, lipid droplet deposition, macrophage infiltration, and inflammation in white adipose tissues, while reducing brown adipose thermogenesis [10,11,12,13,17,18,19,21,25]. Hepatic lipogenesis and impaired lipid metabolism are other targets of olanzapine action [8,20,21,22,38,39]. The hypothalamus, liver, and adipose tissues represent critical central and peripheral organs for the actions of insulin. Among the actions of insulin, Akt plays a key role in integrating metabolic signals, while AMPK/ACC shows additional effects in regulating glucose and lipid homeostasis [22,31,32,33]. Herein, we discovered that olanzapine-induced weight gain and metabolic disturbance were accompanied by decreased hepatic Akt and AMPK/ACC phosphorylation, as well as increased serum IL-6, along with their reversal by chromium. Central activation of AMPK in the hypothalamus increases hepatic glucose production and hyperglycemia [8,40]. In contrast, the peripheral administration of AMPK analog AICAR and hepatic activation of AMPK show improvement in olanzapine-induced hyperglycemia and hyperlipidemia [22,41]. This is to say that the effects of AMPK vary and show tissue difference. However, the actions of hepatic Akt and AMPK/ACC are to maintain homeostatic regulation of glucose and lipid metabolism.

Independent of the central or peripheral effects, anti-oxidation, anti-inflammation, anti-obesity, anti-adiposity, and improvement in glucose and lipid metabolism were all demonstrated in the beneficial treatments against olanzapine [20,21,22,25,28]. Interesting findings which came as a result of this study were that olanzapine treatment caused a depletion of tissue chromium, along with the reversal of olanzapine-induced weight gain and metabolic disturbance from daily chromium supplementation. Chromium is an important nutrient required for both optimal insulin activity and normal carbohydrate and lipid metabolism [29,42]. Its pharmacological activities also include anti-oxidation, anti-inflammation, insulin sensitizer, and AMPK activator [23,31,32,33]. Evidence indicates that chromium loss may be a sign that there’s a defect in glucose metabolism [43]. Additionally, chromium deficiency or lower circulating levels of chromium occurs in diabetic patients and aggravates hyperglycemia, hyperinsulinemia, insulin resistance, hyperlipidemia, adiposity, and obesity [43,44]. Previously, we revealed that daily supplements involving chromium-containing milk powder elevated serum chromium levels and improved glucose and lipid metabolism along with disease severity, in obese-, bile duct ligated-, high-fat diet-fed-, and stroke-mice [23,31,32,33]. The same administrative protocols and dosages also increased tissue chromium levels in olanzapine medicated rats, and improved weight gain and metabolic disturbance. Together with other relevant studies, our results suggest that chromium loss might play an active role in the development of olanzapine-induced weight gain and metabolic disturbance, where chromium supplementation is able to improve olanzapine-accompanied adverse effects.

Despite certain novel findings, the present study has limitations in its interpretation of the precise action mechanisms of chromium, and the interplay between olanzapine and chromium depletion. Stress hormones have been implicated in promoting chromium excretion leading to chromium loss [45]. However, a reduction of stress hormones after its administration [46] primarily excludes their involvement in olanzapine-induced chromium depletion. The crosstalk and underlying mechanisms between olanzapine and chromium depletion are yet to be defined. Control of one’s body weight is directly linked to the balance between food intake and energy expenditure. Chromium appears to have a negligible effect on olanzapine-increased food intake, indicating there’s a possibility of increased energy expenditure as an explanation for the improvement in weight gain. Increasing evidence indicates that an increased expression of UCP-1 in white adipose tissues represents an alternative mechanism for the improvement of metabolic impairment via energy expenditure [35]. Here, we found that chromium not only induced UCP-1 expression in white adipose tissues, but also further promoted that in olanzapine-medicated rats. Although the promoting mechanism of UCP-1 expression in the combinatory treatment of olanzapine and chromium remained unresolved, current findings suggest that an increased thermogenic UCP-1 in white adipose tissues could be a cause for the resolution of weight gain, metabolic disturbance, and inflammation. However, the exact mechanisms of thermogenesis after chromium treatment warrant further investigation. Hyperglycemia and hyperlipidemia could be two consequences of obesity, while olanzapine may have direct effects on the liver and/or adipose tissues, impacting glucose and lipid metabolism. In addition to the various livers examined during the current study, the changes and potential roles of the hypothalamus, adipose tissues, and skeletal muscles represent alternative and valuable targets for the elucidation of chromium actions. Relevant studies have measured a detectable level of serum chromium within a range from 30–1000 µg/L in controlled rats [47,48]. According to current findings, the basal levels of serum chromium in rats varied from 100 µg/L to 400 µg/L. Such variability alerts us to pay more consideration for the interpretation of serum chromium levels. The aforementioned concerns should be investigated in any upcoming studies in order to provide completion of whole actions.

In conclusion, olanzapine medicated rats experienced weight gain and adiposity, as well as the development of hyperglycemia, hyperinsulinemia, insulin resistance, and hyperlipidemia. 

An olanzapine-induced metabolic disturbance was accompanied by decreased hepatic Akt and AMPK actions, as well as increased serum IL-6, along with tissue chromium depletion. Results of the present study show that both chromium mobilization and loss may be an alternative mechanism responsible for olanzapine-induced weight gain and metabolic disturbance. In accordance with the reported metabolic improving effects, daily chromium supplementation was also helpful in resolving post-olanzapine weight gain and metabolic disturbance. Although current findings imply that the beneficial effects of chromium against olanzapine-accompanied adverse effects may be mediated by its improvement in hepatic Akt and AMPK action, alternative beneficial mechanisms might be existed, and would therefore warrant further detailed study.

## 4. Materials and Methods

### 4.1. Ethics Statement

The protocol of this study had been reviewed and approved by the Institutional Animal Care and Use Committee of the Taichung Veterans General Hospital and National Chung Hsing University (IACUC Approval No: 104-004, 23/01/2015). All experimental procedures held to the Guidelines for the Care and Use of Laboratory Animals recommended by the Taiwanese government.

### 4.2. Animal Experiments

The female Sprague-Dawley rats (160 ± 10 g), were obtained from BioLasco, Taiwan Co., Ltd. (Taipei, Taiwan) and housed in a standard animal facility of 22 ± 2 °C, 60% humidity, with a 12-h light/dark cycle. Rats were fed with a regular chow diet (5001 Rodent LabDiet, Bourn Feed & Supply, SE Columobia, MO, USA). During the course of experiments, the rats were allowed free access to food and water. Following one week of habituation, the rats were divided into control and olanzapine groups treating with or without chromium, respectively (*n* = 10 per group). At the end of experiments, rats were euthanized by pentobarbital anesthesia followed by decapitation.

### 4.3. Drug Treatment

Olanzapine was administrated 2 mg/kg via oral gavage twice daily (AM 9:00 and PM 5:00) for two weeks after a one-week acceptance training. Control rats were administered with a corresponding vehicle at the equivalent volume. One hour after the first daily olanzapine administration, the chromium groups were orally supplemented with chromium-containing milk (trivalent chromium dose: 80 µg/kg) made from chromium-containing milk concentrate capsule. The milk concentrate capsule contains 325 ppm trivalent chromium, with a proportion of chromium chloride hexahydrate, lactoferrin, whey protein concentrate, and milk powder (1:6:200:393) provided by Maxluck Biotechnology Corporation, Taipei, Taiwan [33]. All dosages and administrations of olanzapine and chromium followed previously reported studies with slight modifications [23,25].

### 4.4. Biochemical Analyses

The rats were fasted for glucose measurements. The fasting blood glucose was taken with glucose oxidase reaction. After fasting blood glucose were taken, the rats were euthanized, and the organs and blood were taken for samples. Serum triglycerides, total cholesterols, Aspirate aminotransferase (AST), and Alanine aminotransferase (ALT) were measured by clinical chemistry analyzers (Hitachi Autoanalyzer 7070; Roche I 800, Tokyo, Japan). Liver tissues were dissolved in chloroform/methanol (2:1) buffer and analyzed using commercialized assay kits (BioVision; Milpitas, CA, USA) for the measurement of hepatic triglycerides and cholesterol [33]. The levels of insulin (Crystal Chem. Inc.; Elk Grove Village, IL, USA) and IL-6 (R&D Systems, Minneapolis, MN, USA) were determined with enzyme immunosorbent assay (ELISA) kits, following the procedures provided by the manufacturers.

### 4.5. Homeostasis Model Assessment Index

Insulin resistance was evaluated from fasting insulin and glucose levels according to the Homeostasis Model Assessment (HOMA) Insulin Resistance index (HOMA-IR) as follows: HOMA-IR = fasting insulin (μU/mL) × fasting glucose (mmol/L)/22.5 [49]. 

### 4.6. Trace Element Analysis

The resected liver, gastrocnemius, white adipose tissues (including inguinal, periovarian, mesentery, and perirenal fat), serum, and 24-h urine samples removed from the euthanized rats were digested by 65% nitric acid solution, and heated at 65 °C for 1 h [23]. After digestion, the quantifications of chromium in the samples were carried out using graphite furnace atomic absorption spectrophotometry (Hitachi Z-2000 series polarized Zeeman atomic absorption spectrophotometer, Tokyo, Japan).

### 4.7. Histological Examination

The excised liver and perirenal fat of the experimental rats were fixed in a 10% formalin solution and embedded with paraffin. Hematoxylin and Eosin (H&E) staining was performed according to standard procedures [33].

### 4.8. Tissue Preparation and Western Blot Analysis

The excised liver tissues or white adipose tissues (including inguinal, periovarian, mesentery, and perirenal fat) were homogenized with tissue protein reagents (T-PER, Pierce Biotechnology, Rockford, IL, USA) and equally quantified before western blotting. Proteins were separated by electrophoresis with SDS-PAGE and then transferred to PVDF membranes. The membranes were sequentially incubated with 5% non-fat milk, recognizing antibodies, and horseradish peroxidase-labeled IgG. Antibodies recognizing phosphorylated Akt, Akt, phosphorylated AMP-activated Protein Kinase (AMPK), AMPK, phosphorylated Acetyl-CoA Carboxylase (ACC), ACC, uncoupling protein-1 (UCP-1), and β-actin were used (Santa Cruz Biotechnology, Santa Cruz, CA, USA). Lastly, the blots were developed using enhanced chemiluminescence (ECL) substrates with autoradiograph, and quantified by optical densitometry (Image Master ID, Pharmacia Biotech, Upsalla, Sweden).

### 4.9. Statistical Analyses

Results were presented as mean ± standard deviation, with the significance between groups analyzed by one-way analysis of variance (ANOVA) method using the SPSS software. The statistical difference with significance was defined as *p* < 0.05 for all tests.

## Figures and Tables

**Figure 1 ijms-20-00640-f001:**
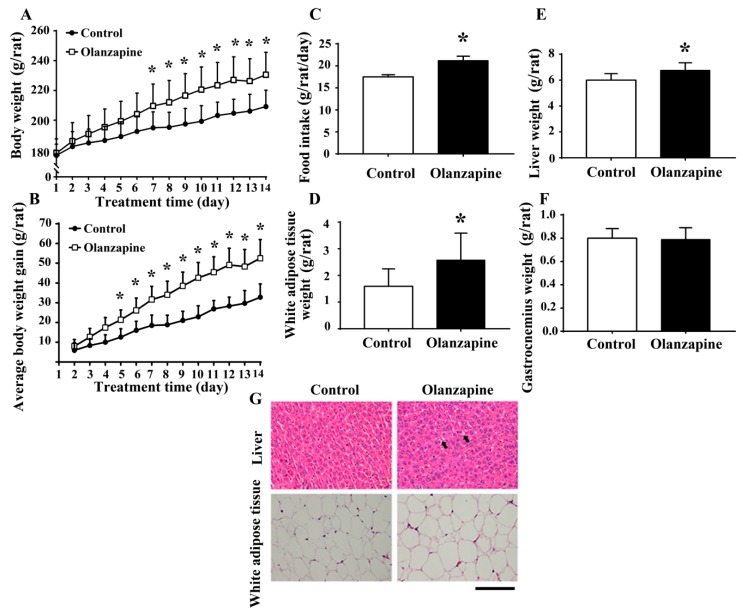
Olanzapine caused weight gain. During a course of 14 days, body weight and food intake were recorded over time. Daily changes in body weight (**A**), average body weight gain (**B**), and food intake (**C**) were summarized and depicted. At the end of the experiments, the weight of excised white adipose tissues (**D**), liver (**E**), and gastrocnemius (**F**) was measured. (**G**) The excised liver and perirenal white adipose tissues were subjected to histological examination. Representative photomicrographs of Hematoxylin and Eosin (H&E) stain are shown. * *p* < 0.05 vs. control, *n* = 10. Scale Bar: 0.1 mm. Arrows indicate a deposition of lipid droplets.

**Figure 2 ijms-20-00640-f002:**
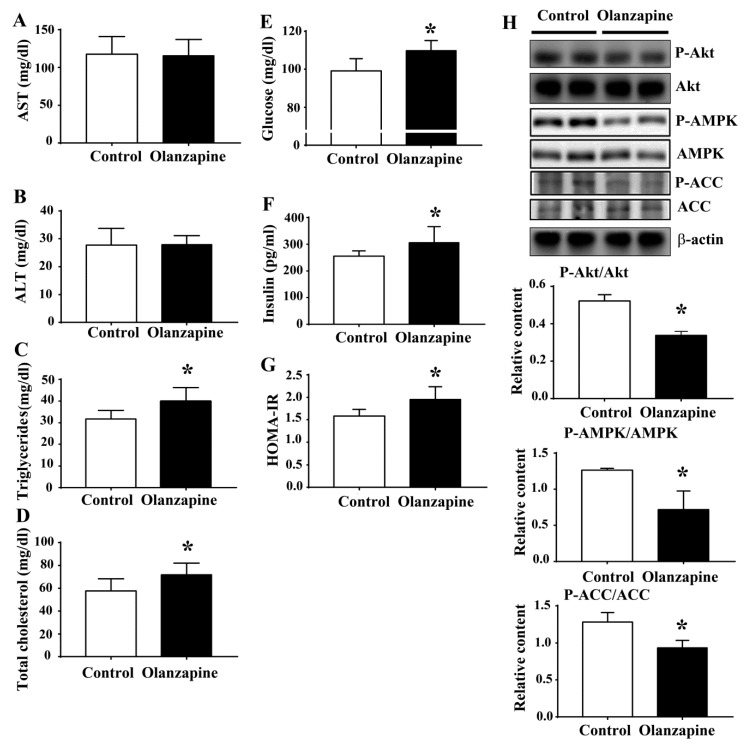
Olanzapine caused a metabolic disturbance. At the end of the experiments, serum samples were subjected to analyses for the measurement of aspartate aminotransferase (AST) (**A**), alanine aminotransferase (ALT) (**B**), triglycerides (**C**), total cholesterol (**D**), fasting glucose (**E**), and insulin (**F**). The value of Homeostasis Model Assessment-Insulin Resistance (HOMA-IR) is depicted in (**G**). (**H**) Proteins were extracted from the excised liver tissues and subjected to Western blotting with indicated antibodies. Representative blots are shown, and the quantitative data are depicted. * *p* < 0.05 vs. control, *n* = 10.

**Figure 3 ijms-20-00640-f003:**
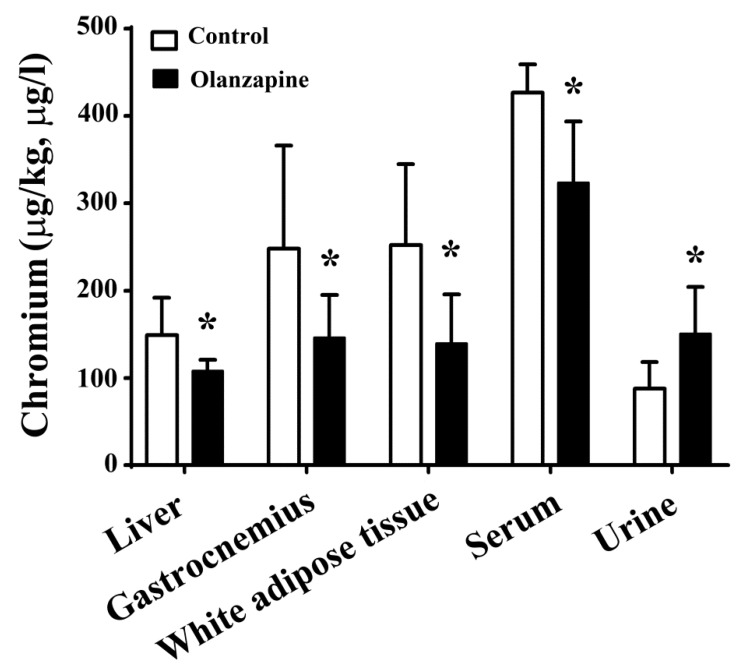
Olanzapine caused tissue chromium mobilization. At the end of the experiments, the excised liver (µg/kg), gastrocnemius (µg/kg), white adipose tissues (including inguinal, periovarian, mesentery, and perirenal fat) (µg/kg), serum (µg/L), and 24-h urine (µg/L) samples were subjected to analyses for the measurement of chromium content. * *p* < 0.05 vs. control, *n* = 10.

**Figure 4 ijms-20-00640-f004:**
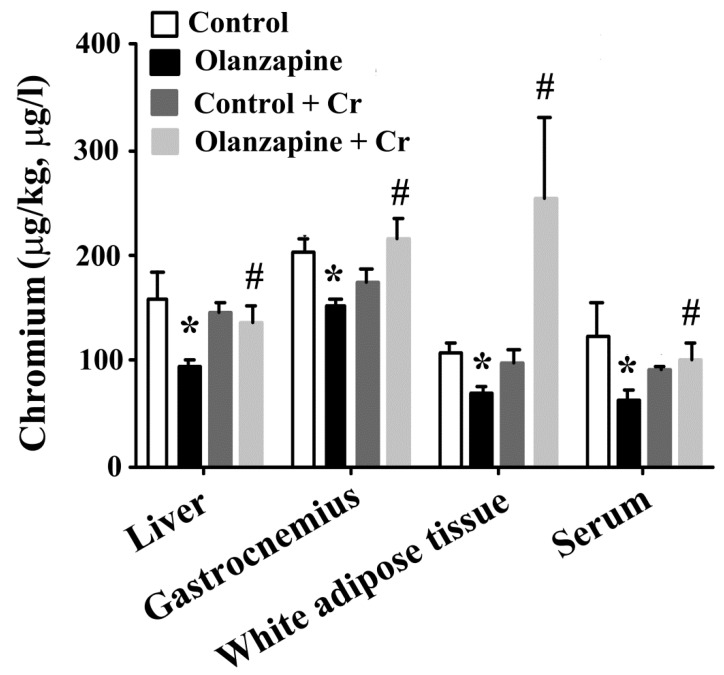
Chromium improved olanzapine-induced tissue chromium mobilization. At the end of the experiments, the excised liver (µg/kg), gastrocnemius (µg/kg), white adipose tissues (including inguinal, periovarian, mesentery, and perirenal fat) (µg/kg), and serum (µg/L) samples were subjected to analyses for the measurement of chromium content. * *p* < 0.05 vs. control and # *p* < 0.05 vs. olanzapine control, *n* = 10.

**Figure 5 ijms-20-00640-f005:**
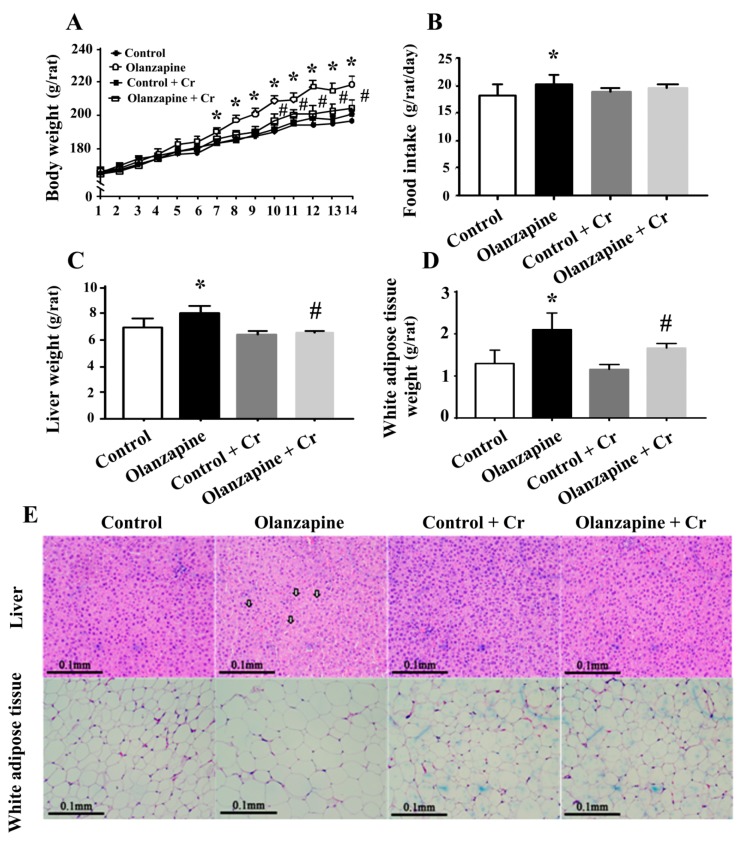
Chromium improved olanzapine-induced weight gain. During a course of 14 days, body weight and food intake were recorded over time. Daily changes in body weight (**A**) and food intake (**B**) were summarized and depicted. At the end of the experiments, the weight of the excised liver (**C**) and white adipose tissues (**D**) was measured. (**E**) The excised liver and perirenal white adipose tissues were subjected to histological examination. Representative photomicrographs of H&E stain are shown. * *p* < 0.05 vs. control and # *p* < 0.05 vs. olanzapine control, *n* = 10. Scale Bar: 0.1 mm. Arrows indicate a deposition of lipid droplets.

**Figure 6 ijms-20-00640-f006:**
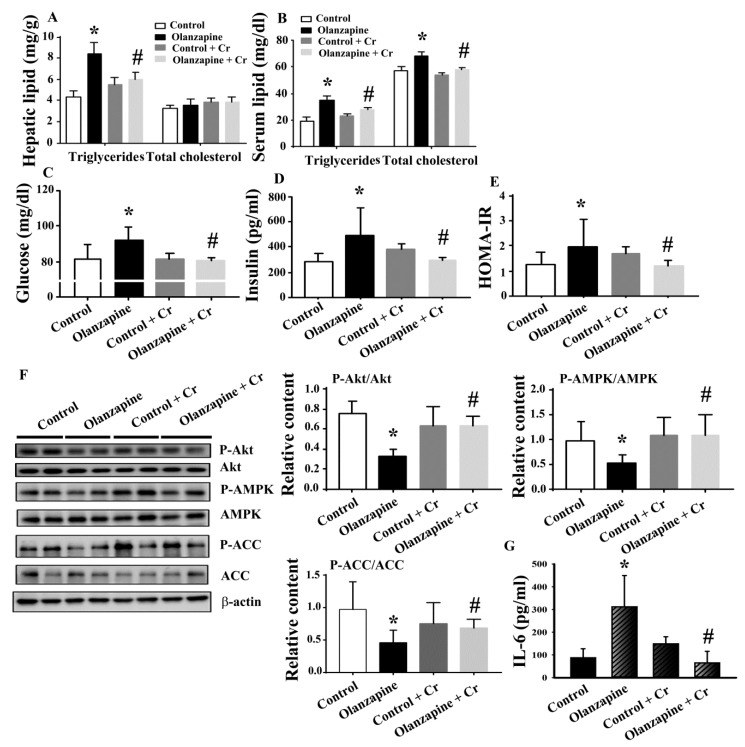
Chromium improved olanzapine-induced metabolic disturbance. At the end of the experiments, tissue lipids were homogenized and extracted from livers and subjected to an enzymatic assay for the measurement of hepatic triglycerides and cholesterol (**A**). Serum samples were subjected to analyses for the measurement of lipids (**B**), fasting glucose (**C**), and insulin (**D**). The value of HOMA-IR is depicted in (**E**). (**F**) Proteins were extracted from the excised liver tissues and subjected to Western blotting with indicated antibodies. Representative blots are shown, and the quantitative data are depicted. (**G**) Serum samples were subjected to analyses for the measurement of IL-6. * *p* < 0.05 vs. control and # *p* < 0.05 vs. olanzapine control, *n* = 10.

**Figure 7 ijms-20-00640-f007:**
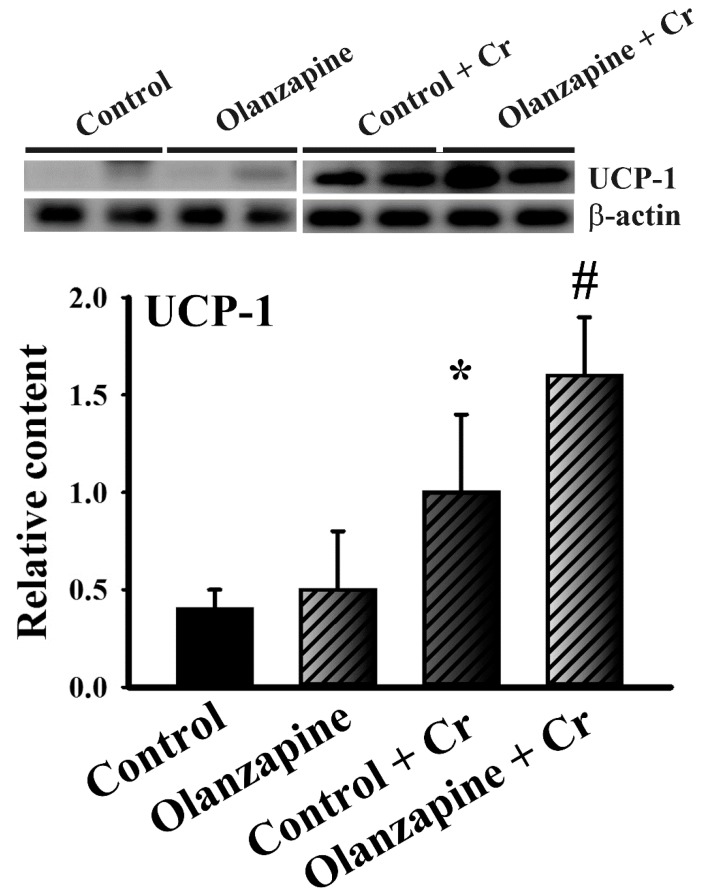
Chromium increased uncoupling protein-1 (UCP-1) expression in white adipose tissues. Proteins were extracted from the excised white adipose tissues (including inguinal, periovarian, mesentery, and perirenal fat) and subjected to Western blotting with indicated antibodies. Two representative blots of 10 rats were shown and the quantitative data were depicted. * *p* < 0.05 vs. control and # *p* < 0.05 vs. olanzapine control, *n* = 10.

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
