# Peer review of "Olanzapine Induced Dysmetabolic Changes Involving Tissue Chromium Mobilization in Female Rats"

_ijms, 2019, doi:10.3390/ijms20030640_

Round 1
Reviewer 1 Report
This submitted manuscript deals with the mitigation of the adverse endocrinometabolic effects of olanzapine treatment in rats obtained by dietary supplementation with chromium-containing milk concentrate capsules. The authors clearly indicate and interpret that such supplementation is attenuating the weight gain, fat deposition, liver steatosis and insulin resistance induced by olanzapine treatments. This rat model is well accepted to resume the widely recognized adverse effects of olanzapine when used as a treatment of schizophrenic patients. Moreover the authors bring evidence that olanzapine treatment on its own disturbs/reduces the chromium availability in healthy rats. The beneficial actions of chromium supplementation therefore suggest to propose the use of chromium as a co-treatment for olanzapine therapy, in order to increase the ratio of benefits/risks of this antipsychotic drug, recognized to treat schizophrenia but to trigger metabolic syndrome. However, besides the clear demonstration of chromium utility in this model of drug-induced dysmetabolism, there are several points that require clearer information before publication. The following concerns should be concisely treated/added in an already well-written manuscript without absolute need of supplementary experiments
1) Apparently, the authors have performed two sets of experiments: the former to confirm the well-known metabolic disturbances of olanzapine treatment in female rats and to evidence a kind of chromium deficiency provoked by the antipsychotic drug, the latter to demonstrate the benefits of chromium supplementation. However, while showing in Figs 3 and 4 that olanzapine administration reduces serum chromium levels in a similar manner in these two experiments, the authors did not discuss the large inter-experiment variations in the levels of chromium in the serum of control rats (around 4-fold change). Thus, the authors have to propose reasons for this difference, or they add such variability as a limitation of their studies/interpretations, as well as evoking at least once putative Chromium toxicity.
2) There is more than a simple addition of the effects of Cr and olanzapine on UCP-1 protein content in adipose tissue. Such synergism looks strange alongside the diverse other effects that are apparently opposite between Cr and olanzapine: bw, fattening, liver injury… This aspect should be more mentioned in Discussion. Additionally it is unclear what white adipose tissue(s) has(ve) been studied since singular/plural is used alternatively throughout the text, and line 328 is not informative enough about this concern. Lastly, an adipose tissue that expresses UCP1 is at least beige/brite or brown but not typically white. Consequently, modifications of lines 34-35, 110, 196, and of other occurrences are welcome.
3) A gender difference is generally admitted regarding the obesogenic or diabetogenic adverse effects of olanzapine, explaining that the authors have treated groups of 10 female rats. However, nothing is told about gender differences and beneficial effects of chromium. A sentence in discussion should clarify both aspects.
Minor defects requiring correction during revision:
1) The style of the title could be improved since "Chromium supplements improved …" appears rather confusing at the first glance. As far the referee understands, there is only one type of Cr supplement studied; and it is not the supplement that is improved but rather the dysmetabolism of olanzapine "medicated" rats, written here as " bw gain & metabolic disturbance", afar from the term "improved". Other combination of words might be more attractive …
2) line 66 What is meant by ", deposition of white adipocytes "? fat deposition ?
3) all y axes of figures 1 & 5 international unit of grams is "g" and not "gm" . Verify whether also applies for ppb in fig 3 & 4 or define in Methods or in figure legend.
4) line 117 "triglyceride" should be replaced by "serum triglycerides" . "Triglycerides" should also be used in L 125, 151…
5) line 128 & 185: the use of "representative blots ARE shown" will be highly appreciated.
6) line 133-4: use "chromium levels … have decreased , " instead.
7) line 211 Is " Cr homeostasis, with a net loss occurring through urinary excretion" meant ?
8) line 253 "beyond basic nutrition " ?
9) lines 257, 275, 281: unclear, please rephrase.
10) line 300: is 2 mg/kg bw /day meant ?
11) lines 322, 328, 332: " at the end of expts , rats were euthanized" looks redundant.
Author Response
First, we thank deeply to those constructive and instructive comments from the reviewers and editors. We sincerely considered these comments and made an appropriate improvement of this revised manuscript (ijms-431828). We had provided graphic abstract and tried our best to edit and correct the revised version of manuscript with an aim to decrease similarity index. The detailed summary of the changes in relation to the reviewers was listed as follows.
1) Responses: In this study, two sets of rats showed variable levels of serum chromium from 100 to 400 mg/l. According to relevant studies, basal levels of serum chromium were detected within a range from 30-1000 mg/l. Therefore, we should pay more attention in explaining current findings. We had added a description in the section of Discussion.
Please refer to line 282-286.
2) Responses: UCP-1 is attracting focus for its role in metabolism. Its upregulation and activation are demonstrated to have beneficial roles in metabolic diseases, including obesity. It is not a typical molecule in white adipose tissues. Instead, its expression represents a change of beige/brite adipocytes. Current findings identified its upregulation in white adipose tissues following chromium treatment. Intriguingly, chromium further promoted UCP-1 expression in olanzapine-medicated rats. We had added description and discussion regarding the findings to the revised manuscript.
Pleases refer to line 192-197, 270-277.
3) Responses: The adverse effects of olanzapine are well-known of gender difference. However, data of clinical and experimental findings imply that the biological activity of chromium is working without gender difference. To simplify the study model, only female SD rats were enrolled. A brief description was added.
Please refer to line 210-213.
Minor defects requiring correction during revision:
1) Responses: As suggested by the reviewers, the title was changed to “Olanzapine induced dysmetabolic changes involving tissue chromium mobilization in female rats”.
2) Responses: As suggested by the reviewers, the phrase was changed to fat deposition. Please refer to line 64-68.
3) Responses: As suggested by the reviewers, the units were changed to g/rat and g/rat/day. The unit of ppb was changed to mg/kg (liver, gastrocnemius, adipose tissues) or mg/l (serum, urine).
4) Responses: As suggested by the reviewers, triglyceride was changed to triglycerides.
5) Responses: As suggested by the reviewers, the sentence was changed to “Representative blots are shown and the quantitative data are depicted.”.
6) Responses: As suggested by the reviewers, the sentence was changed. Please refer to line 134-136.
7) Responses: As suggested by the reviewers, the sentence was changed.Please refer to line 218-220.
8) Responses: As suggested by the reviewers, the sentence was changed by removing the three words.
9) Responses: As suggested by the reviewers, the sentences were changed.
10) Responses: As suggested by the reviewers, the description of doses was changed.
11) Responses: As suggested by the reviewers, the redundant sentences were removed.
Reviewer 2 Report
The manuscript presents interesting data, the study was well designed and described and I suggest only minor revision.
There are some issues in this manuscript:
1. In Fig 1 and Fig 5 Author used unit: gm/rat or gm/rat/day it should be explained or may be change (g/rat ?);
2. In method section, Author wrote that rats were euthanized-what method? It is important because euthanasia methods may affect the results.
Author Response
1. Responses: As suggested by the reviewers, the units were changed to g/rat and g/rat/day.
2. Responses: As suggested by the reviewers, the protocol of animal euthanasia was described. “At the end of experiments, rats were euthanized by pentobarbital anesthesia followed by decapitation.”.